# The Role of Vitamin D and Its Molecular Bases in Insulin Resistance, Diabetes, Metabolic Syndrome, and Cardiovascular Disease: State of the Art

**DOI:** 10.3390/ijms242015485

**Published:** 2023-10-23

**Authors:** Christiano Argano, Luigi Mirarchi, Simona Amodeo, Valentina Orlando, Alessandra Torres, Salvatore Corrao

**Affiliations:** 1Department of Internal Medicine, National Relevance and High Specialization Hospital Trust ARNAS Civico Di Cristina Benfratelli, 90127 Palermo, Italy; luigi.mirarchi@arnascivico.it (L.M.); simona.amodeo@arnascivico.it (S.A.); valentina.orlando03@community.unipa.it (V.O.); alessandra.torres1993@gmail.com (A.T.); salvatore.corrao@unipa.it (S.C.); 2Department of Health Promotion Sciences, Maternal and Infant Care, Internal Medicine and Medical Specialties, [PROMISE], University of Palermo, 90127 Palermo, Italy

**Keywords:** vitamin D, insulin resistance, metabolic syndrome, type 1 and 2 diabetes, gestational diabetes, cardiovascular diseases and metabolism

## Abstract

In the last decade, an increasing awareness was directed to the role of Vitamin D in non-skeletal and preventive roles for chronic diseases. Vitamin D is an essential hormone in regulating calcium/phosphorous balance and in the pathogenesis of inflammation, insulin resistance, and obesity. The main forms of vitamin D, Cholecalciferol (Vitamin D3) and Ergocalciferol (Vitamin D2) are converted into the active form (1,25-dihydroxyvitamin D) thanks to two hydroxylations in the liver, kidney, pancreas, and immune cells. Some anti-inflammatory cytokines are produced at higher levels by vitamin D, while some pro-inflammatory cytokines are released at lower levels. Toll-Like Receptor (TLR) expression is increased, and a pro-inflammatory state is also linked to low levels of vitamin D. Regardless of how it affects inflammation, various pathways suggest that vitamin D directly improves insulin sensitivity and secretion. The level of vitamin D in the body may change the ratio of pro- to anti-inflammatory cytokines, which would impact insulin action, lipid metabolism, and the development and function of adipose tissue. Many studies have demonstrated an inverse relationship between vitamin D concentrations and pro-inflammatory markers, insulin resistance, glucose intolerance, metabolic syndrome, obesity, and cardiovascular disease. It is interesting to note that several long-term studies also revealed an inverse correlation between vitamin D levels and the occurrence of diabetes mellitus. Vitamin D supplementation in people has controversial effects. While some studies demonstrated improvements in insulin sensitivity, glucose, and lipid metabolism, others revealed no significant effect on glycemic homeostasis and inflammation. This review aims to provide insight into the molecular basis of the relationship between vitamin D, insulin resistance, metabolic syndrome, type 1 and 2 diabetes, gestational diabetes, and cardiovascular diseases.

## 1. Introduction

In recent years, attention to the role of vitamin D in different fields is growing. Vitamin D is a liposoluble prohormone with endocrine, autocrine, and paracrine functions and is fundamental to bone metabolism [1]. Vitamin D has a role in extra-skeletal functions; consequentially, there is a relationship between vitamin D deficiency and some pathologic conditions, including diabetes, metabolic syndrome, non-alcoholic liver disease, autoimmune diseases, hypertension, cardiovascular disease, and cancer [2,3,4,5,6,7,8,9] (Figure 1). Moreover, the recent pandemic of COVID-19 has underlined the possible therapeutic role of Vitamin D in some aspects of the infection and the association between severe vitamin D deficiency and COVID-19-related health outcomes [10,11,12]. Many studies have reported the existence of immuno-modulatory effects of vitamin D and that its deficiency may be associated with a sub-inflammatory state [13]. Diabetes and metabolic syndrome represent a major clinical and public health problem. The disease burden related to diabetes and metabolic syndrome is increasing significantly, particularly in older subjects [14,15]. According to the International Diabetes Federation, data released in 2021 showed that 537 million adults live with diabetes worldwide. The total number is predicted to rise to 643 million by 2030 and to 783 million by 2045 instead of the previous estimation of 693 million [16]. Many epidemiological and observational studies have found an association between vitamin D insufficiency and the incidence of type 1 and type 2 Diabetes [17,18,19,20,21]. In this sense, many studies reported the existence of different mechanisms able to explain the potential role of vitamin D in glucose metabolism, such as the preservation of the β-cell function and slow failure of residual β-cell function in patients with type 1 diabetes and latent autoimmune diabetes [22,23]. Furthermore, vitamin D determines direct stimulation of insulin secretion and improves peripheral insulin resistance by reducing systemic inflammation via the vitamin D receptor on pancreatic beta cells and in muscles and the liver [24,25,26]. This last mechanism also plays a key role in metabolic syndrome development [27]. The lack of vitamin D receptors in cardiovascular tissue increased ventricular mass dysregulation of metalloproteinases and fibroblasts, promoting the fibrotic process and ventricular dilatation [28].

Given this background, an extensive search of SCOPUS, PubMed, and CENTRAL was performed using the following string ((vitamin D) or (calcifediol)) or (ergocholecalciferol)) AND (systematic review [pt] or meta-analysis [pt]) and 2017:2023 [dp]). The search string retrieved 1575 manuscripts. The hand-searching of principal generalist, human nutrition, and basic research journals was carried out as well. Two authors (V.O. and A.T.) independently reviewed the titles, abstracts, and full texts of the retrieved articles to determine their potential inclusion. Any disagreements were resolved via discussion with a third author (S.C.). Manuscripts regarding the role of vitamin D in insulin resistance, type 1 and 2 diabetes, gestational diabetes, metabolic syndrome, and cardiovascular disease were extracted for this review. This review aims to explore the molecular basis of the role of vitamin D in insulin resistance, type 1 and 2 diabetes, gestational diabetes, metabolic syndrome, and cardiovascular disease.

## 2. Vitamin D Metabolism

Vitamin D is a liposoluble prohormone that humans can acquire via nutrition and synthesis in the skin during exposure to UV radiation [29]. Vitamin D3 (Cholecalciferol) is the main source of vitamin D, and vitamin D2 (Ergocalciferol) is the form through which vitamin D exists. 

Most of the amount of Cholecalciferol comes from the endogenous production in the skin after sun exposure; a small amount of Cholecalciferol has an exogenous origin and derives from foods. Ergocalciferol is contained in dairy products and nutritional supplements, and it is a non-animal form of vitamin D [30].

Once in circulation, Cholecalciferol and Ergocalciferol are converted in the liver tissue by the action of vitamin D-25-hydroxylase (CYP2R1) to 25-hydroxyvitamin D or calcifediol [25 (OH) D]; subsequently, 25(OH)D undergoes a second conversion, by the enzyme 25-hydroxyvitamin D-1α-hydroxylase (CYP27B1), into active and bioavailable vitamin D (1,25-dihydroxyvitamin or calcitriol—CT) [1,25 (OH)2 D] [31,32,33]. This reaction takes place mainly in the kidney. At that point, 1,25 (OH)2 D performs its functions by binding to the vitamin D receptor (VDR), expressed in the cytoplasm of cells, forming a VDR-RXR hormone complex (vitamin D receptor—retinoid X receptor) via the stimulation of the heterodimerization of the VDR with the retinoid X receptor [15]. In the nucleus, it regulates the expression of many genes via their up or downregulation [32]. 1,25 (OH)2 D has about 1000-fold higher affinity than 25(OH) for the VDR. CYP27B1 is also expressed in other tissues, like activated macrophages, microglia, parathyroid glands, breast, colon, and keratinocytes; 1,25 (OH)2 D has autocrine and paracrine effects [34,35] (Figure 2). It is known that vitamin D is associated with bone health and can play an essential role in other systems, including the immune system. These extra skeletal actions are available because of the presence of VDR and hydroxylation enzymes in different tissues such as the pancreas, kidney, muscles, liver, and others. Vitamin D supplementation (VDS) has hormonal, anti-inflammatory, anti-apoptotic, anti-fibrotic activities, antioxidant, and immune-modulatory effects [36,37], and also plays a role in insulin resistance via the reduction in the expression of some pro-inflammatory cytokines like interleukin-1 (IL-1) and IL-6 [38].

## 3. Vitamin D and Insulin Resistance

### 3.1. Vitamin D, Insulin Resistance, and Molecular Mechanisms

Vitamin D is involved in several non-skeletal health diseases, including common metabolic disorders like Metabolic Syndrome (MetS), Type 2 Diabetes (T2DM), Impaired Fasting Glucose (IFG), Non-Alcoholic Fatty Liver Disease (NAFLD), and Polycystic ovarian syndrome (PCOS), which are all characterized by insulin resistance (IR) [39,40,41]. It has been demonstrated that there is an inverse association between vitamin D deficiency and the Homeostatic Model Assessment of Insulin Resistance (HOMA-IR), which is used as the measure of insulin resistance and defined as an increase in insulin secretion necessary for the maintenance of glycemic homeostasis [42]. Therefore, the supplementation of vitamin D reduces the risk of insulin resistance and circulating levels of insulin [42,43]; the inverse correlation between vitamin D and HOMA-IR becomes more robust with increasing Body Mass Index (BMI) [42]. 

Molecular mechanisms underlying the pathophysiological hypothesis of the possible association between hypovitaminosis D and insulin resistance are mainly associated with the expression of insulin receptors, and the production of inflammatory cytokines and polymorphism of VDR expressed in the β-cells of the pancreas. In particular, vitamin D acts upon gene transcription via genomic and non-genomic mechanisms. Based on the above, hypovitaminosis D and insulin resistance are genetically interrelated [40,42,43]. 

Concerning insulin receptor expression, it was found that vitamin D increases receptor expression in muscle, liver, and adipose tissue, improving insulin sensitivity [42]. In detail, it was shown that vitamin D works as an epigenetic factor, affecting the transcription level of many genes involved in insulin sensitivity, like Insulin Receptor Substrate (IRS), which is increased by 2.4-fold in high-fat mice models treated with vitamin D [42]. As a result, insulin sensitivity improves in the target tissues because IRS protein increases insulin sensitivity [42]. In addition, vitamin D improves the sensitivity of insulin receptors to insulin and glucose transport and promotes the conversion of proinsulin to insulin [43,44,45].

Vitamin D deficiency increases the expression of pro-inflammatory cytokines, which can be the cause of insulin resistance in patients with relatively higher BMI; it has been observed that obesity is associated with hypovitaminosis D because of three reasons: less exposure to sunlight, the low intake of vitamin D via nutrition, and the sequestration of vitamin D in the adipose tissue [42]. In addition to this mechanism, it was found that high secretion of the anti-diabetic hormone leptin, whose levels are deregulated by abdominal adiposity, is associated with insulin resistance. This means that high doses of vitamin D supplements can decrease leptin levels and reduce BMI in insulin-resistant patients [42]. This effect would be linked to a reduced caloric intake mediated by the binding of vitamin D to its receptors in the paraventricular nucleus of the hypothalamus.

As regards VDR, it is an endocrine member of the nuclear receptor superfamily for steroid hormones, and it works as a transcription factor that mediates the action of vitamin D via the control of the expression of hormone-sensitive genes, like Calmodulin-Dependent Kinase (CaMKs), which in turn stimulates VDR-mediated transcription by phosphorylation levels of VDR [40]. The function of β-cells may be affected by vitamin D via direct and indirect mechanisms. The direct mechanism consists of binding of vitamin D to VDR in β-cells, helping in the release of insulin secretion [42,46]; the indirect mechanism is related to the regulation by vitamin D of calcium flux via the pancreatic β-cell because insulin secretion is strongly dependent on calcium [46]. This could be the reason why tissue calcium levels (adipose tissue and skeletal muscle) affect IR [46]. It was recently discovered that the deletion of macrophage VDR promotes insulin resistance [40]. 

Recently, it was found that the enzyme-activating vitamin D, 1-α-hydroxylase, is present in β-cells [46]. While the non-genomic functions of vitamin D are carried out via the activation of numerous signaling molecules (phosphatidylinositol-3 kinase, phospholipase C (PLC), Ca2+-calmodulin kinase II (CaMPKII), protein kinase A (PKA), mitogen-activated protein kinases (MAPK)2s2+, src, and protein kinase C (PKC)) that in turn interact with vitamin D response elements (VDRE) on the promoter of vitamin D-sensitive genes [42,47]. In addition to those mentioned, there are other mechanisms by which vitamin D, via an alternative non-genomic pathway, influences intracellular signaling molecules or transcription factors that condition the expression of various genes. 

This action could explain some of the modulatory effects of vitamin D on innate and adaptive immunity, cell antiviral responses, and cell survival. This pathway involves the protein–protein interaction between the VDR and a target protein. Some of these target proteins are represented by kinase I-κB (IKK)β, one of the upstream regulators of the NF-κB canonical pathway [33], signal transducers and transcription activators (Stat)1, Runt-related transcription factor (RunX) 1 [35], c-jun, β-catenin, and cAMP response element binding protein [48]. 

### 3.2. Studies and Research

A recent meta-analysis, which included 9232 participants, has studied genetic associations of four polymorphisms in the VDR with insulin-resistant diseases, particularly TaqI, BsmI, ApaI, and FokI variants. It was found that there is an association between insulin resistance-related diseases (mostly with PCOS and MetS than T2DM) and the VDR ApaI variant (mostly G allele than T allele) in Asians and populations who lived in middle-latitude districts. The BsmI (mostly A allele than G allele) and TaqI variants (T/C allele) were more prevalent in dark-pigmented Caucasians. At the same time, there was no association between the VDR FokI variant and insulin resistance-related diseases in populations with different skin pigments and in different latitudes [40]. 

Beneficial effects of high-dose vitamin D (≥2000 mg/day) and calcium (≥1000 mg/day) in both short-term and long-term (>12 weeks) combined vitamin D and calcium supplementation were found [46]. However, the results obtained so far are conflictive because some trials reported that the supplementation of vitamin D does not reduce insulin resistance [29]. Further studies, like long-term and large-scale randomized controlled trials, are needed.

## 4. Vitamin D and Type 2 Diabetes Mellitus (T2DM) 

### 4.1. Vitamin D, T2DM and Molecular Mechanisms

It is well known that T2DM is a public health challenge worldwide, accounting for approximately 87–91% of all cases of diabetes. Type 2 Diabetes Mellitus is a chronic metabolic disorder characterized by inadequate insulin production and consequentially high blood glucose [49]. T2DM constitutes an essential risk factor for premature death and adverse complications, micro and macrovascular, such as blindness, stroke, heart attack, amputation, and kidney failure [50], and also determines and impairs quality of life [51]. According to OMS, 762 million people worldwide suffer from pre-diabetes [52], which is strongly connected with obesity [53]. 

A lack or insufficiency of Vitamin D is associated with macrovascular and microvascular complications of T2DM [54]. Obesity, prediabetes, and T2DM are often characterized by low circulating vitamin D levels [55]. 

VDR, implicated in the systemic effect of vitamin D, is also expressed in high insulin-sensitive tissues (pancreas, adipose tissue, and muscle) [56]. In the body, vitamin D is an epigenetic factor mediating the transcription level and enhancing insulin sensitivity [42]. 

### 4.2. Studies and Research

According to a recent meta-analysis, vitamin D supplementation improves glycemic homeostasis and insulin sensitivity [57]. It also seems to work as an anti-diabetic factor by regulating insulin sensitivity and production, controlling parathyroid hormone levels, and anti-inflammatory cytokine effects [58,59]. Vitamin D has been identified as a potential prevention and treatment strategy [60]. Low 25-hydroxyvitamin D (25(OH) D) levels are highly prevalent among T2DM patients [61]. The effects of vitamin D supplementation may explain the association between vitamin D and T2DM because it prevented the increase in plasma HbA1c levels and in IR [62,63,64]. 

The gold standard for evaluating glycemic control in T2DM is represented by glycated hemoglobin (HbA1c) in line with the UK Prospective Diabetes Study [65]. 

In line with recent studies, vitamin D supplementation is implicated in plasma HbA1c reduction, suggesting that vitamin D can contribute to reducing the development of diabetic complications [66]. Also, studies have found that vitamin D supplementation improved beta cell function [67] and insulin sensitivity [56,68,69,70], especially in those at high risk for diabetes.

In particular, vitamin D has a role in lipid metabolism in adipose tissue [71] and may decrease inflammation [72]. In pancreatic tissue, Vitamin D protects β-cells function, reducing local inflammation [73,74]. A key role is represented by the activation of the VDR expressed in the pancreatic beta-cell. Indeed, mice lacking VDR have impaired insulin secretion [75], and the addition of Vitamin D stimulates pancreatic cells, resulting in increased insulin secretion [76]. It is worth outlining that the human insulin receptor gene promoter contains a Vitamin D response element, suggesting that transcriptional activation of the gene may be favored by calcitriol administration [77,78]. A calcium-dependent mechanism mediates insulin secretion. Vitamin D may play a role [79] in regulating the opening and closure of calcium channels, mediating the calcium flux in beta cells, and interacting with receptors (VDR and 1,25 D3-MARRS). Therefore, vitamin D deficiency causing an alteration in calcium flux could interfere with normal insulin secretion [80,81]. In addition, vitamin D is involved in skeletal muscle metabolism, insulin sensitivity, and lipid composition [82]. Consequently, increasing circulating vitamin D concentration could affect tissue energy and metabolism, improving systemic insulin sensitivity. The skeletal muscle is crucial in insulin sensitivity, involving 70–90% of total glucose disposal during the post-prandial period [83,84,85]. Thus, vitamin D supplementation might improve skeletal muscle glucose handling and, as a consequence, insulin sensitivity [86]. Vitamin D also regulates the adipose tissue, and hypovitaminosis may play a role in obesity and fat mass due to the restoration of Vitamin D, a fat-soluble vitamin, in the adipose tissue [87]. According to Bajaj et al., hypovitaminosis also seems to increase microvascular complications such as diabetes retinopathy, diabetic neuropathy, diabetic nephropathy, and diabetic foot ulcers [88], and a meta-analysis demonstrated that increased circulating vitamin D levels protect the kidney from injury and ameliorate proteinuria in T2DM patients [89]. Concerning microvascular complications, vitamin D deficiency may be involved in diabetic neuropathy interfering with nociceptor functions by causing diabetic nerve damage [90], and diabetic retinopathy increasing the severity and playing a role in the pathogenesis via its effects on the immune system and angiogenesis [91]. Lastly, a lack of Vitamin D promotes macrovascular complications such as endothelial dysfunction and arterial stiffness [92,93], peripheral arterial disease, and carotid arterial plaque [94]. Vitamin D might have a direct effect on vascular stiffness. Vascular smooth muscles (VSMCs) and endothelial cells express 1α-hydrolase, which is involved in the conversion of 25(OH)D to calcitriol [95]. It has been shown that this enzyme is activated in primary cultures of human umbilical vein endothelial cells by inflammatory molecules such as TNF-α and lipopolysaccharide [96]. In addition, it was found that vitamin D has a direct effect on vascular tone by reducing calcium influx [97]. Moreover, an extrarenal activation of vitamin D was suggested as a possible contributor to hypertension and arterial stiffness [98]. 

The vitamin D level is inversely related to blood pressure [99]. Experimental and human studies showed that subjects with 1,25(OH)_2_D_3_ deficiency have increased activity of the RAAS, both in the body and in the kidney, developing hypertension [100]. An increased plasma renin concentration and low 1,25(OH)_2_D_3_ levels may elevate sympathetic activity and enhance intra-glomerular pressure, predisposing to AH, a decline in GFR, and subsequent cardiovascular damage [101]. The knocking out of either the VDR or the 1α-hydroxylase gene in mice upregulates RAAS activity and induces HT [102,103], while treatment of these animals with 1,25(OH)_2_D_3_ suppresses the RAAS activity [103]. In addition, the VDR is expressed in vascular tissues, including the myocardium, renin-producing juxtaglomerular cells, and vascular smooth muscle, which directly influences calcium influx, muscle relaxation, and diastolic function [104,105]. 

Hence, hypovitaminosis D (as deficiency or insufficiency) embraces several complications in diabetic patients; therefore, screening for vitamin D levels in T2DM patients may play a crucial role in defining the outcomes. 

## 5. Vitamin D and Type 1 Diabetes Mellitus (T1DM)

### 5.1. Vitamin D, T1DM and Molecular Mechanisms

Type 1 Diabetes Mellitus (T1DM) is a chronic autoimmune disease related to an immune system alteration that destroys pancreatic ß cells with a consequent quantitative or qualitative dysfunction of insulin [106]. The prevalence of T1DM has steadily increased over the past few decades in most countries [107]. Patients with T1DM are genetically susceptible to developing autoimmune diseases, with an increased risk of developing the disease among first-degree relatives [108,109]. Currently, the research aims to identify genetic and environmental factors predisposing to the onset of the disease. Current knowledge suggests that an important role could be played by vitamin D, which in the first years of life modulates the still-growing immune system, which plays a crucial role in the development of self-tolerance [110,111,112]. Vitamin D signaling impairment, especially in the first years of life, increases the risk of autoimmunity [113,114,115]. Given the role that vitamin D plays in the immune system, it is believed that it may have a protective role in the development of T1DM [116]. 

The discovery of vitamin D receptors throughout the body has opened up new reflections on its possible implication in other diseases, including autoimmune diseases such as T1DM and multiple sclerosis [117]. Indeed, VDR is also expressed in immune cells, effectively regulating innate and adaptive immune responses [118]. 

The expression of 1α-hydroxylase CYP27B1 in specific immune system cells explains how these can regulate vitamin D levels [119]. Some studies have shown that the activity of macrophages/monocytes, antigen pre-transmitter cells, T cells, and B cells is regulated by vitamin D [120]. It plays a role in the modulation of the activity of dendritic cells [121]. In the presence of 1,25(OH)2D3, dendritic cells produce fewer inflammatory factors such as tumor necrosis factor-α and interleukin-12, producing a less anti-inflammatory tolerance state characterized by increased production of interleukin-10 [121].

1,25(OH)2D3 promotes macrophage differentiation, which is essential for the activation of involutional inflammation in animal models of T1DM, to the anti-inflammatory phenotype (M1→M2) via the VDR- PPARgamma signaling pathway [122]. These properties of vitamin D on the regulation of the inflammatory response are very interesting in T1DM because, in the pancreas of affected patients, there is an inflammatory infiltrate composed of T lymphocytes, B lymphocytes, and macrophages. In animal models of T1DM, such as non-obese diabetic mice, high doses of calcitriol and non-high calcium vitamin D analogs arrest involutional inflammation, as indicated by reduced effector T cell numbers and the induction of T-reg cells [123,124,125].

Fronczak et al. reported that increased maternal intake of vitamin D in food reduced the risk of autoimmunity against pancreatic beta cells in their offspring; there is no effect of 1alpha,25-dihydroxyvitamin D3 on residual beta cell function and insulin requirements in adults [126]. 

### 5.2. Studies and Research

According to evidence (systematic reviews and meta-analyses) [127,128] on the link between vitamin D levels and T1DM, adequate vitamin D status in the first years of life reduces the risk of diabetes [17,110,112,129], and vitamin D deficiency is more common in people with T1DM [130,131]. A cross-sectional study revealed that 70% of children with T1DM had a vitamin D deficiency [132], and rickets are associated with an increased risk of T1DM [133]. Also, the TEDDY study reported that a higher infant concentration of 25(OH)D is associated with lower islet autoimmunity [134]. In contrast, a birth cohort study in Finland suggested that sufficient vitamin D supplementation could assist in decreasing T1DM risk [133].

The risk of developing T1DM before age 15 is associated with a reduction in serum vitamin D levels, as demonstrated by a case–control study that was part of EURODIAB (OR 0.63) [135]. Human studies report the relationship between VDR polymorphisms and T1DM risk and β cell function. Although 25D is the major circulating form, pancreatic β cells can convert 25D to 1,25D [136]. This implies that a small role in beta cell survival in T1DM can be played by exogenous and circulating 1,25 D. Anyhow, rising 25D levels could be helpful as a substrate for the formation of 1,25D by beta cells while circulating 1,25 D could exert autocrine and paracrine effects. Considering beta cell injury at the clinical diagnosis, vitamin D is much less likely to be helpful after disease onset [137]. A meta-analysis conducted by Najjar et al. found no critical effect of a genetically determined reduction in 25(OH)D concentrations by selected polymorphisms on T1DM risk. A meta-analysis conducted by Gregoriou et al. showed that vitamin D supplementation in patients with T1DM resulted in a reduction in daily insulin requirements, as well as improving fasting C-peptide (FCP), stimulated C-peptide (SCP), and HbA1c [138].

Although not all studies agree [139], there are many observational studies that show a strong association between vitamin D deficiency and T1DM [140].

## 6. Vitamin D and Gestational Diabetes Mellitus (GDM)

### 6.1. Pathophysiology of Vitamin D Levels in Pregnancy

In pregnancy, numerous physiological alterations of the maternal metabolism are necessary for the normal development of the fetus. During pregnancy, a relationship between the maternal and fetal vitamin D status underlines the importance of an adequate vitamin D level in this period. Gestational vitamin D metabolism adaptations include a characteristic physiological growing of 1,25(OH)2D in maternal blood. It rises at the beginning of gestation and reaches its highest levels in the third trimester, where it presents two to three times the levels found in non-pregnant women. Several studies have shown a correlation between vitamin D levels and GDM [141]. GDM is defined as glucose intolerance, and IR was first diagnosed in pregnant women [142]. GDM affects up to 14% of pregnancies [143]. Inadequate glycemic control in women with GDM leads to short- and long-term maternal complications, including gestational hypertension, preeclampsia, macrosomia, congenital abnormalities, hypoglycemia in the newborn, and an increased risk of T2DM after pregnancy [144,145]. GDM is compared to a form of impaired glucose tolerance, similar to prediabetes in non-pregnant individuals, and represents a global public health problem related to serious health problems in the mother and newborn [146]. Women with a history of GDM have an increased risk of developing IR syndrome (IRS) and cardiovascular disease (CVD) later in their lives [147]. The rate of women who develop T2DM within 5–10 years ranges from 20 to 60% [148,149]. The risks of occurrence of MetS and CVD are three times higher in women with GDM. Indeed, children born to women with GDM have a higher risk of developing impaired glucose tolerance and obesity. The pathogenesis of GDM has not yet been cleared. Some studies [150,151] suggest that the onset and development of GDM are closely related to genetic factors (insulin resistance, family history of diabetes, and immune dysfunction) and environmental factors (dietary structure and pancreatic β cell damage). 

### 6.2. Vitamin D, GDM, and Molecular Mechanisms

Vitamin D can support insulin secretion and normal glucose tolerance [152]. Vitamin D deficiency seems closely related to the onset of GDM. Among the factors that may play a role in the onset of GDM is chronic low-grade inflammation [153]. The increased degree of inflammation in early pregnancy is related to an increased risk of GDM and the development of hyperglycemia [154]. Moreover, in women with GDM, oxidative stress has been found [155,156,157], while antioxidant status is downregulated [158]. Oxidative stress plays an important role in both the pathogenesis and complications of GDM [159]. A significant inverse association exists between serum vitamin D concentrations and low-grade inflammation [160]. The low levels of vitamin D trigger inflammatory responses via the NF-kB pathway by regulating p-p65/RelB in pancreas tissue [161] upwards. Excessive Ca2+ and reactive oxygen species (ROS) in ß cells, both in vitamin D deficiency, result in cell death and promote diabetes [162].

Furthermore, some genes that protect against the onset of diabetes are inactivated by hypermethylation [163]. Vitamin D prevents hypermethylation by increasing the expression of DNA demethylases in more regions of genes that protect against diabetes [162]. In addition, a significant inverse association was also found between serum calcium concentrations, which is positively regulated by vitamin D, and obesity risk as another diabetes risk factor [164].

### 6.3. Studies and Research

Several longitudinal prospective cohort studies have reported the risk of GDM with serum vitamin D concentrations in early pregnancy [165]. A meta-analysis conducted by Chunfeng Wu et al. [166] showed that vitamin D supplementation has a beneficial effect on lipidic assessment: it increases HDL-Cholesterol (HDL-C) levels and is useful for reducing serum Total Cholesterol (TC) and LDL-Cholesterol (LDL-C) levels of patients with GDM. However, no single opinion exists between this meta-analysis and the previous ones [167,168,169].

Preceding meta-analyses [168] pointed out that vitamin D can improve LDL-C levels but does not affect triglycerides (TG), TC, and HDL-C. The short duration of the studies could explain this. Several studies [170,171,172] have shown that when GDM patients have abnormal lipid metabolism, their risk of pregnancy complications increases. Studies [173,174] proved that vitamin D deficiency is associated with a higher incidence of T2DM and vitamin D supplementation can dramatically increase insulin sensitivity in people with IR and vitamin D deficiency. IR and insufficient secretion underlie the pathogenesis of GDM [175]. According to a network meta-analysis conducted by Shixiao Jin et al. to evaluate the effects of vitamin D supplementation, it was best for reducing fast plasma glucose (FPG) and improving HOMA-IR compared to the effects of other nutritional strategies [176]. Vitamin D deficiency is a frequent phenomenon after pregnancy; one study showed that at 25–28 weeks of gestation, the concentration of 25(OH)D (the active form of the vitamin within the body) in GDM patients is significantly reduced [177]. Another systematic review and meta-analysis conducted by Wang M. et al. showed how vitamin D supplementation in a population of women with GDM can statistically significantly reduce serum FPG, insulin, and HOMA-IR, as well as complications related to childbirth (cesarean section, maternal hospitalization, and postpartum hemorrhage) and newborns (hyperbilirubinemia, giant children, hypoglycemia, polyhydramnios, fetal distress, and premature delivery). Vitamin D deficiency is considered a potential risk factor for abnormal glucose metabolism; Zhang et al. [178] conducted a study that showed that low vitamin D levels in the blood may increase the risk of GDM and that adequate vitamin D supplementation may improve GDM status. 25(OH)D can not only regulate insulin secretion but also stimulate insulin receptor expression to promote insulin sensitivity [179], achieving the effect of lowering blood sugar. In addition, vitamin D has antioxidant effects, which can reduce β islet cell damage and apoptosis β of islet cells via active oxidative groups [180]. Patients with GDM can increase their 25(OH)D concentration via vitamin D supplementation, thereby improving insulin resistance and decreasing blood sugar [181].

## 7. Vitamin D, Metabolic Syndrome (MetS), and Cardiovascular Disease (CVD)

### 7.1. MetS and CVD: Burden of the Problem

MetS is related to abdominal obesity, IR, hypertension, and dyslipidemia [182]. The diagnosis of MetS includes waist circumference (WC), FPG, TG levels, HDL-C levels, total cholesterol levels, and blood pressure (BP) [183]. The MetS increases the risk of developing T2DM associated with long-term microvascular and macrovascular damage [184] and CVDs. CVDs are one of the significant causes of disability and death worldwide [185]. Atherosclerosis is the primary etiology of CVDs, and it is considered a chronic inflammatory condition [186]. Several studies have also documented that a decrease in antioxidant levels and an increase in inflammatory and oxidative stress biomarkers may be involved in the pathophysiology of T2DM complications [187] and the onset of CVDs [188]. The inflammatory process can be triggered by metabolic disorders such as atherogenic dyslipidemia (higher TG and apolipoprotein B, small low-density lipoprotein cholesterol LDL-C particles, and low HDL-C concentrations), T2DM, and increased inflammatory cytokines [189]. Consequently, the inflammatory cascade may initiate plaque formation, endothelial damage, and, ultimately, plaque rupture [186]. The pathophysiology of endothelial dysfunction includes the overproduction of reactive oxidative species, inflammatory cytokines, pro-atherogenic lipoproteins, and an imbalance between vasodilating and vasoconstricting molecules. The impairment of vasodilation may be due to reduced bioavailability of nitric oxide (NO), produced by the endothelial cells and involved in multiple physiological processes, including vasodilation, inflammation, and platelet aggregation [190]. On the other hand, dyslipidemia is associated with insulin resistance and elevated risk of CVD events [191,192]. There are numerous risk factors for MetS e CVDs; among these, the dietary factor is among the most important [193], such as high-calorie and high-fat diets [194]. 

### 7.2. Vitamin D, MetS, CVD, and Molecular Mechanisms

Vitamin D deficiency patients are a risk factor for MetS [195]. Vitamin D deficiency can affect insulin secretion and sensitivity and play an essential role in the onset of MetS [27]. Furthermore, a study found that vitamin D supplementation had a positive effect on lipid profile, IR, hyperglycemia, obesity, and hypertension and then on the treatment of MetS-related disorders [196].

Vitamin D can reduce Oxidative Stress (OS) using upregulating cellular Glutathione (GSH) and antioxidant systems such as glutathione peroxidase and superoxide dismutase [197]. Also, vitamin D can inhibit Reactive Oxygen Species (ROS) secretion [198]. VDRs are expressed in different tissues, notably endothelial cells, vascular smooth muscle cells, and cardiomyocytes, and regulate the expression of the target gene [199]. Vitamin D3, furthermore, is a direct transcriptional regulator of endothelial Nitric Oxide (NO) synthase. In this pathophysiological situation, OS plays a crucial role in cellular injury, in which the production of reactive ROS suppresses the antioxidant defense system of the cells, which consequently causes cellular death [200]. Under physiologic conditions, the antioxidant defense systems maintain the oxidant-antioxidant balance by adjusting the altering levels of oxidants [201]. The antioxidant defense systems include enzymes such as glutathione peroxidase, catalase, superoxide dismutase, and other compounds (albumin and GSH).

Furthermore, different nutrients such as vitamins and minerals can also affect the antioxidant balance [202,203,204]. Accordingly, vitamin D has been proposed to have antioxidant properties. The association between VDS and MetS is controversial. The benefits of VDS in the treatments of MetS and its disorders connected include improved arterial stiffness, mitochondrial oxidation, and phospholipid metabolism; increased lipoprotein lipase activity, peripheral insulin sensitivity, and β-cell function; and decreased inflammatory cytokines and parathyroid hormone levels and renin–angiotensin–aldosterone system activity [205,206,207,208].

Various underlying mechanisms have been suggested for the association of serum vitamin D with MetS and its components. Firstly, there is an inverse significant relation between blood vitamin D concentration and abdominal obesity.

Vitamin D is a fat-soluble vitamin and tends to be stored in adipose tissues; so, its bioavailability and circulating levels are lower in those with abdominal obesity, so the synthesis in the liver of obese individuals is lower compared to ordinary people [87]. Considering the central role of vitamin D in the expression of insulin receptors and increasing insulin responsiveness for glucose transporters (GLUTs), serum vitamin D deficiency is involved in the incidence of insulin resistance and type 2 diabetes [77]. Also, vitamin D is known as an antihypertensive agent because of its direct effect on vascular cells, suppression of the renin–angiotensin–aldosterone system, calcium metabolism, and prevention of secondary hyperparathyroidism [209], in particular, parathyroid hormone (PTH), which is also involved in the process of lipogenesis; Vitamin D deficiency can favor greater adiposity by promoting an increase in parathyroid hormone levels and the inflow of calcium in adipocytes, thus increasing lipogenesis and inhibiting the lipolysis in adipocytes [210].

Another hypothesis suggests that excess body fat retains the metabolites of this vitamin, and cholecalciferol produced in the skin or acquired via the diet is partially “sequestered” by body fat before being transported to the liver and undergoing the first hydroxylation [211]. 

Obese persons may have low levels of 25(OH)D in their blood due to adipose tissue impeding vitamin D absorption and usage or because they spend less time outside, leading to insufficient vitamin D production in the skin [212]. Other proposed mechanisms include high expressions of the vitamin D receptor in adipose tissue and the possibility of vitamin D playing a role in the pathogenesis of the metabolic syndrome [213]. The benefits of Vitamin D supplements in the treatments of MetS and its disorders connected include improved arterial stiffness, mitochondrial oxidation, and phospholipid metabolism; increased lipoprotein lipase activity, peripheral insulin sensitivity, and β-cell function; and decreased inflammatory cytokines and parathyroid hormone levels and renin–angiotensin–aldosterone system activity [205,206,207]. In conclusion, there is no clear and demonstrated relationship between low serum 25(OH)D levels and MetS because of the lack of long-term studies.

### 7.3. Studies and Research

Zhu and Heil reported that serum 25D level was linked to the risk factors for MetS [214]. In a meta-analysis study, Jafari et al. [215] reported that vitamin D supplementation improved the lipid profile of patients with T2DM. In another meta-analysis, vitamin D intake significantly decreased insulin resistance in people with T2DM [216]. Several RCTs have studied the impact of vitamin D supplements on lipid profiles, glucose homeostasis, and C-reactive protein (CRP) in persons with CVDs [217]. Some studies reported no significant relationship between VDS and MetS in adults [218,219,220]. Therefore, the association between VDS and MetS still needs evidence to demonstrate whether VDS helps treat MetS. In a meta-analysis, Ostadmohammadi, Milajerdi et al. demonstrated the beneficial effects of vitamin D supplementation on reductions in fasting glucose, insulin concentrations, and HOMA-IR. In addition, the pooled analysis revealed a significant increase in serum HDL-C concentrations after vitamin D therapy and a significant reduction in CRP levels. However, supplementation did not affect TG, TC, and LDL-C levels [221]. Kai-Jie, Zhong-Tao et al. conducted a meta-analysis to study the effect of vitamin D on MetS in adults using relevant biomarkers such as anthropometric parameters, BP, blood lipid profile, blood sugar, OS, and vitamin D toxicity. Vitamin D did not affect waist circumference, body mass index, body fat percentage, and BP. VDS significantly reduced FPG but did not affect HDL-C, LDL-C, TC, and TG blood levels. For OS parameters, VDS significantly lowered malondialdehyde and hypersensitive CRP [222]. De Paula et al. [223], in a systematic review, meta-analysis, and randomized clinical trials (RCTs), investigated the effects of micronutrients on BP in patients with T2DM. In this systematic review, a reduction in BP, especially systolic BP, was demonstrated. Observational and experimental data favor the concept that vitamin D is associated with the pathogenesis of arterial hypertension [224,225]. A possible mechanism for this link involves the inhibition of the renin–angiotensin–aldosterone system by vitamin D. Additionally, in the presence of hypovitaminosis D, an alternative mechanism could be related to the secondary hyperparathyroidism and relative hypocalcemia that are commonly seen in these patients [226]. In a meta-analysis, Hajhashemy Z, Shahdadian F et al. illustrated that the highest level of blood vitamin D, compared with the lowest level, was significantly linked to lower odds of MetS in cross-sectional studies on the adult population. In addition, based on dose–response analysis, each 25 nmol/L (or 10 ng/mL) increment in 25(OH)D was associated with a 15% decreased chance of MetS.

## 8. Discussion and Conclusions

Data reported in our review support the notion that Vitamin D levels are associated with T1DM and T2DM, GDM, MetS, and CVDs. There is some experimental and epidemiological evidence for the administration of Vitamin D in these different diseases. In Table 1, we have summarized the results of meta-analyses and systematic reviews on the effectiveness of vitamin D administration and their dosages in the various conditions that we have dealt with in this review. However, data from randomized clinical trials, which are highly heterogeneous, yielded contrasting data. Although the possibility of preventing the onset of the disease, vitamin D administration should be started very early in life or even during pregnancy in T1DM and GDM; moreover, different data showed that vitamin D administration improves glucose metabolism and the risk of T2DM and metabolic syndrome, randomized clinical studies showed contradictory results for vitamin D supplementation in the management of altered metabolic states. In this sense, further studies are necessary to determine the fundamental role of vitamin D deficiency and if it can be considered a causal factor in altered metabolism. 

## Figures and Tables

**Figure 1 ijms-24-15485-f001:**
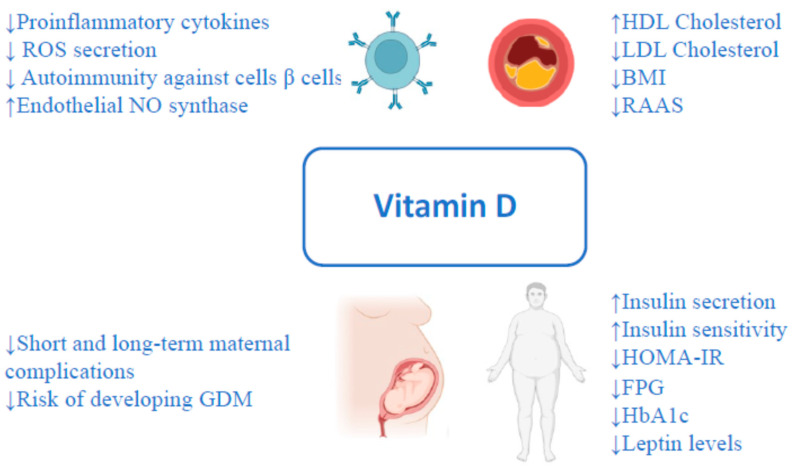
The pleiotropic effect of vitamin D.

**Figure 2 ijms-24-15485-f002:**
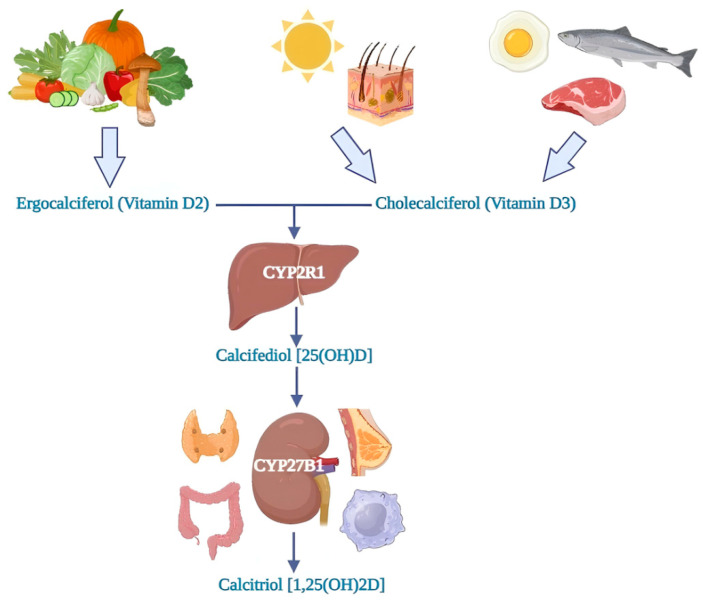
Vitamin D metabolism.

**Table 1 ijms-24-15485-t001:** Synthesis of meta-analyses and systematic reviews on the effectiveness of various dosages of vitamin D administration in pathologic conditions analyzed.

Author/Year	Design	Duration	Participants (I/C)	Dose of Vitamin D	Results
VITAMIN D AND INSULIN RESISTANCE (IR)					
Asbaghi et al., 2019 [46]	MT (12 RCTs)	From 6 to 312 weeks	8946 healthy subjects or patients with overweight/obesity, IFG, pre-diabetes, GDM, T2DM, PCOS, HIV infection (4395/4551)	From 200 IU/day Vitamin D3 to 50,000 IU/week Vitamin D3 (with supplementation dose of calcium that ranged from 500 mg/day to 1000 mg/day)	Reduce effects on FBG, circulating levels of insulin, and HOMA-IR
Sindhughosa et al., 2022 [41]	MT (7 RCTs)	From 10 to 52 weeks	735 patients with NAFLD (423/312)	From 1000 IU/day Vitamin D3 to 50,000 IU/week Vitamin D3	Improvement in IR (marked by a decrease in HOMA-IR), decrease in ALT levels
Pienkowska et al., 2023 [29]	SR (8 RCTs)	From 12 to 260 weeks	From 66 to 2423 patients with prediabetes	From 1000 IU/day Vitamin D3 to 88,000 IU/week Vitamin D3	Only one trial showed improvements in FBG and HOMA-IR
VITAMIN D AND TYPE 2 DIABETES MELLITUS (T2DM)					
Pittas et al., 2007 [61]	MT (13 case–control studies; 15 cross-sectional studies; 12 RCTs)	N/A	Patients with T2DM or prediabetes	2000 IU/day Vitamin D3 or Vitamin D3 700 IU/day with supplementation dose of 500 mg/day calcium citrate	Vitamin D and calcium insufficiency may negatively influence glycemia, whereas combined supplementation with both nutrients may be beneficial in optimizing glucose metabolism
Krul-Poel et al., 2017 [86]	MT (23 RCTs)	From 4 to 52 weeks	1797 patients with T2DM: for the effect on HbA1c 1475 patients (755/720), for the effect on FBG 1180 patients (608/572)	From 1000 IU/day Vitamin D3 to 45,000 IU/week Vitamina D3 or 11,200 IU/day Vitamin D3 for 2 weeks followed by 5600 IU/day for 10 weeks or from 100,000 to 300,000 IU Vitamin D3 single dose	Significant effect on FBG in a subgroup of studies (n = 4); no significant effect in change in HbA1c
Mirhosseini et al., 2018 [57]	MT (28 RCTs)	From 8 to 260 weeks	3848 healthy subjects or patients with prediabetes and/or overweight or obesity, NAFLD, arterial hypertension, cervical intraepithelial neoplasia, premenopausal and postmenopausal women	From 420 IU/day to 88,880 IU/week Vitamin D3	Significant reduction in HbA1c, FBG, and HOMA-IR
Hu et al., 2019 [66]	MT (19 RCTs)	From 4 to 24 weeks	1374 patients with T2DM (747/627)	Up to 50,000 UI/weekly Vitamin D3 or 300,000 UI single injection Vitamin D3	Significant reduction in HbA1c, IR (marked by a decrease in HOMA-IR) and insulin levels in the short-term vitamin D supplementation group
VITAMIN D AND TYPE 1 DIABETES MELLITUS (T1DM)					
Gregoriou et al., 2017 [138]	MT (7 RCTs)	From 4 to 52 weeks	287 patients with T1DM	Calcitriol 0.25 μg per day or on alternate days plus insulin Alphacalcidole 0.5 μg daily plus insulin Cholecalciferol 2000 IU per day plus insulin for 18 mo Cholecalciferol 70 IU/kg body weight/day plus insulin	Vitamin D supplementation in the form of alphacalcidole and chole- calciferol appears to be beneficial in daily insulin dose (DID), fasting C- peptide (FCP), stimulated C-peptide (SCP), and HbA1c.
Najjar et al., 2021 [140]	MT (10 studies: 3 cohort; 5 case–control; 2 matched case–control)	N/A	39,884 patients with T1DM (16,370/23,514)	N/A	No large effect of a genetically determined reduction in 25(OH)D concentrations by selected polymorphisms on T1D risk
Hou et al., 2021 [129]	MT (16 studies: 12 case–control studies; 1 cross-sectional case–control study; 2 nested case–control study; 1 case–cohort study)	N/A	10,605 patients with T1DM (3913/6692)	N/A	Results demonstrated a significant inverse association between the 25(OH)D concentration in circulation and the risk of T1DM
Nascimento et al., 2022 [139]	SR (10 studies)	From 6 to 52 weeks	Children and adolescents (0–19 years) with T1DM	Cholecalciferol, with dosages ranging from 1000 to 160,000 IU. Just one study used vitamin D in the form of alfacalcidol at a dosage of 0.25 to 0.5 μg/day	This study did not provide evidence to support the effect of vitamin D supplementation on glycemic control to aid in the treatment of T1DM
Yu et al., 2022 [137]	SR (13 studies: 9 RCTs; 2 open-label case–control; 1 open label; 1 cohort)	From 4 to 12 weeks	527 patients with T1DM	The following therapeutic regimens were used: 1.25 D 0.25 μg 2nd daily; 25 D 2000 IU daily; 25 D to achieve serum 25 D > 125 nmol/L; Alfacalcidol 0.25 μg bd 25 D; 60,000 IU monthly; Ergocalciferol (D2) 2 m of 50,000 IU/w; 25 D 2000 IU/d; 25D. 3000 IU/d; Calciferol 2000 IU/d + etanercept + GAD-alum	The maintenance of optimal circulating 25 D levels may reduce the risk of T1D and that may have potential for benefits in delaying the development of absolute or near-absolute C-peptide deficiency
VITAMIN D AND GESTATIONAL DIABETES MELLITUS (GDM)					
Akbari et al., 2017 [167]	MT (6 RCTs)	From 6 to 12 weeks	371 pregnant women with GDM (187/184)	50,000 IU vitamin D3 2 times during the study or 50,000 IU vitamin D3 once every 2 weeks for 2 months, for a total of 200,000 IU vitamin D3 or 50,000 U vitamin D3 pearl twice during the study + 1000 mg calcium per day or 1000 IU vitamin D3 and 1000 mg evening primrose oil (EPO) or one intramuscular injection of 300,000 IU of vitamin D3 or a total of 700,000 IU vitamin D3 during pregnancy	This meta-analysis has demonstrated that vitamin D supplementation may lead to an improvement in HOMA-IR, QUICKI, and LDL-cholesterol levels but did not affect FPG, insulin, HbA1c, triglycerides, total-, and HDL-cholesterol levels; however, vitamin D supplementation increased HOMA-B.
Jahanjoo et al., 2018 [169]	MT (5 RCTs)	From 6 to 16 weeks	310 women with GDM	50,000 IU vitamin D3 2 times during the study or 200,000 IU vitamin D3 for each of the first 2 days, and then 50,000 IU per week thereafter, up to 700,000 IU in total. Those at week 28 of gestation or later were asked to take 100,000 IU weekly or 50,000 IU of vitamin D3 once every 2 weeks	This study showed that supplementation of GDM women with vitamin D may lead to an improvement in FPG, TC, LDL, HDL, and hs-CRP serum levels, as well as in newborns’ hyperbilirubinemia
Rodrigues et al., 2019 [170]	MT (6 RCT studies)	From 6 to 24 weeks and a study until delivery	456 pregnant women with GDM diagnosed in the second or third trimester of pregnancy	50,000 IU of vitamin D3 every 2 weeks or 1000 UI daily	Improves adverse maternal and neonatal outcomes related to GDM
Milajerdi et al., 2021 [145]	MT (29 studies: 18 cohort; 9 nested case–control; 1 prospective cross-sectional; 1 retrospective cohort)	N/A	42,668 patients with GDM or not	Blood vitamin D levels	The lowest risk of GDM was found among those with serum vitamin D levels of 40 and 90 nmol/L
Wang et al., 2021 [44]	MT (19 RCTs; of these, 13 concerned GDM)	From 6 to 12 weeks	1198 patients with GDM	From 50,000 IU of vitamin D3 2 times/day to 1200 IU daily	The results showed that vitamin D supplementation during pregnancy could significantly reduce maternal cesarean section rate, maternal hospitalization rate, and postpartum hemorrhage in women with GDM
Chatzakis et al., 2021 [174]	MT (15 studies: 9 cohort; 6 nestedcase–control)	N/A	42,636 pregnant women (1848/40,788)	Blood vitamin D levels	The result showed that lower levels of serum 25(OH)D were associated with a higher chance of GDM
Wu et al., 2023 [166]	MT (20 RCT studies)	From 2 to 16 weeks	1682 pregnant women with GDM diagnosed (837/845)	From 50,000 IU of vitamin D3 2 times/day to 1200 IU daily	Reduce serum LDL-C, TG, and TC levels and increase the serum HDL-C level. Reduce maternal and neonatal hyperbilirubinemia and hospitalization risk.
VITAMIN D, METABOLIC SYNDROME (MetS), AND CARDIOVASCULAR DISEASE (CVD)					
De Paula TP et al., 2017 [223]	MT (7 RCTs)	From 3 to 52 weeks	542 patients with T2DM (472/70)	A single dose of vitamin D2 (100,000 IU) or vitamin D3 (100,000 IU or 200,000 IU)	Reduction in BP, especially in systolic BP
Ostadmohammadi et al., 2019 [221]	MT (8 RCTs)	From 8 to 24 weeks	630 adults with CVD (305/325)	50,000 IU/week Vitamin D3 or 50,000 IU every two weeks or 300,000 IU single dose	Improving glycemic control, HDL-C, and CRP levels; it did not affect TG, TC, and LDL-C levels
Hajhashemy Z. et al., 2021 [226]	Dose–response MT (43 epidemiological studies: 38 cross-sectional; 1 nested case control; 4 cohort studies)	N/A	309.206 adults with or without MetS	Blood Vitamin D levels in adults	Inverse association between serum vitamin D concentrations and risk of MetS
Qi K.J. et al., 2022 [222]	MT (13 RCTs)	From 8 to 24 weeks	1.076 adults with MetS (530/546)	From 1000 IU/day Vitamin D3 to 50,000 IU/week	Decreased BP, FPG, HOMA-IR, and CRP levels; it did not affect HDL-C, LDL-C, TC, and TG levels

I/C: intervention/control; IR: insulin resistance; MT: meta-analysis; RCTs: randomized controlled trials; IFG: impaired fasting glucose; GDM: gestational diabetes mellitus; T2DM: type 2 diabetes mellitus; PCOS: polycystic ovary syndrome; FBG: fasting blood glucose; HOMA-IR: homeostatic model assessment of insulin resistance; NAFLD: non-alcoholic fatty liver disease; HbA1c: glycated hemoglobin; SR: systematic review; IR: insulin resistance; T1DM: type 1 diabetes mellitus; N/A: not applicable; MetS: metabolic syndrome; CVD: cardiovascular disease; BP: blood pressure; TG: triglycerides; TC: total cholesterol; LDL-C: LDL-cholesterol; HDL-C: HDL-cholesterol; CRP: C-reactive protein.

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
