# Peer review of "The Role of Vitamin D and Its Molecular Bases in Insulin Resistance, Diabetes, Metabolic Syndrome, and Cardiovascular Disease: State of the Art"

_ijms, 2023, doi:10.3390/ijms242015485_

Round 1

Reviewer 1 Report

Minor Revision:

Argano et al., review primarily focused on molecular understanding of Vitamin D in regulating type 1 and 2 diabetes, insulin resistance, metabolic syndrome, and cardiovascular diseases.  

Overall, the review is very well presented and covers important aspects underlying the molecular basis of Vitamin D's role in diabetes and other cardiovascular diseases. Here below are my suggestions

1.       Section 2 and Section 3 are overlapping in terms of Vitamin D and insulin sensitivity secretion/b-cell function. I recommend authors avoid this overlap and make it very clear and distinguish how vitamin D regulates insulin resistance and type 2 diabetes mellitus.

2.       Type 2 diabetes is mainly associated with obesity and associated cardiovascular complications including endothelial dysfunction and hypertension. I would recommend authors discuss Vitamin D's role in regulating adiposity, vascular dysfunction, and hypertension and the mechanisms by which it contributes to these complications.

3.     I recommend including the figure for depicting Vitamin D synthesis and metabolism in section 1 (Vitamin D Metabolism)

4.       In Section 4: Vitamin D and Type 1 Diabetes Mellitus: Part of this section was mainly focused on how Vitamin D regulates adaptive and innate immune responses which is nothing to with or relevant to how Vitamin D is regulating Type 1 diabetes. Please exclude this section from the manuscript.

Minor english language changes required in order to improve the manuscript presentation 

Author Response

Dear Reviewer thank you for your valuable comments helpful to improve our manuscript.

According to your suggestion we avoid overlap between vitamin D regulation in insulin resistance and type 2 diabetes. We added a more detailed explanation in "Vitamin D, MetS and CVD and molecular mechanisms section"

According to your suggestion we have discussed vitamin D’s role in regulation adiposity, vascular dysfunction and hypertension in "Vitamin D, MetS and CVD and molecular mechanisms section"

According to your suggestion we included a figure for depicting vitamin D metabolism see figure 2

According to your suggestion in section 4 we exclude the part regarding adaptive and innate immune response.

Reviewer 2 Report

Suggestions for Authors:

1.Figure 1. It is necessary to bring part of the figure to a single type. Absolutely unreadable RAAS fragment. It is not entirely clear what each fragment reflects and how it is related to vitamin D. It is necessary to indicate the effect of vitamin D under each fragment.

2. Sentence "Ergocalciferol is contained in dairy products and nutritional supplements and is the vegetal form of vitamin D [30]" It is not correct. Ergocalciferol  is not vegetal form of vitamin D. It is optimal to name it -  "non-animal form ofvitamin D".

3. There is insufficient information about the non-genomic effect of vitamin D derivatives on the development of specific diseases mentioned in the manuscript.

4. There is no information about the effect of alternative vitamin D derivatives (20-hydroxy or 22-hydroxy, etc.) on the development of specific diseases indicated in the manuscript.

Author Response

Dear Reviewer, thank you for your valuable comments helpful to improve our manuscript.

According to your suggestion we modified figure 1 and we indicate the effect of vitamin D clearly.

Thank you for your observation we corrected the sentence about ergocalciferol.

According to your suggestion we added more information about non -genomic effect of vitamin D.

Thank you for your suggestion about the alternative vitamin d derivates but we narrowed our search only to the terms present in the string

Reviewer 3 Report

1.      Method is too short and must be explained in detail. How many papers were listed after search of listed words. Who performed literature search. Were all papers in the listed period included or some were excluded and why?

2.      It was clearly noted that only meta-analyses and systematic reviews were included in this paper. I was surprised when reading a paper, particularly parts studies and research some other study results were presented. Why table 1 was presented in the paper if results of other studies were reported? Then it cannot be that only MA and SR were included?

3.       A cross-sectional study revealed that 70% of children with T1DM had a vitamin D deficiency [123], and rickets are associated with an increased risk of T1DM [124]. In this sentence there are results of two studies, the first is cross-sectional but the other is birth-cohort. Why these results were reported in the same sentence, jet no one of reported study is not MA or RS, and why their results were reported?

4.      Then it should be revised criteria and reported that not only MA and SR were selected, but other types of studies as well.

Author Response

Dear Reviewer thank you for your valuable comments helpful to improve our manuscript.

According to your suggestion we explained in detail the search method and we reported that not only MA and Sr were selected but other type of studies. See the introduction section.

The search string retrieved 1575 manuscripts. Manuscripts regarding insulin resistance, type1 and type 2 diabetes, gestational diabetes, metabolic syndrome and cardiovascular disease were eligible for this review.

Thank you for your punctual observation. Since in our manuscript we have considered not only meta-analysis and systematic review but also other types of studies, we have reported the cross-sectional study and the birth study as well. We presented table 1 with meta-analysis and systematic reviews because we want to give a synthesize of the evidence available on this topic. 

Round 2

Reviewer 3 Report

The manuscript has been improved and the authors have addressed all my comments.